# Mechanistic Modelling of Slow and Fast NHEJ DNA Repair Pathways Following Radiation for G0/G1 Normal Tissue Cells

**DOI:** 10.3390/cancers13092202

**Published:** 2021-05-03

**Authors:** Yaping Qi, John William Warmenhoven, Nicholas Thomas Henthorn, Samuel Peter Ingram, Xie George Xu, Karen Joy Kirkby, Michael John Merchant

**Affiliations:** 1School of Nuclear Science and Technology, University of Science and Technology of China, Hefei 230026, China; qypzkd@mail.ustc.edu.cn; 2Division of Cancer Sciences, Faculty of Biology, Medicine and Health, University of Manchester, Manchester M13 9PL, UK; john.warmenhoven@manchester.ac.uk (J.W.W.); nicholas.henthorn@manchester.ac.uk (N.T.H.); samuel.ingram@postgrad.manchester.ac.uk (S.P.I.); karen.kirkby@manchester.ac.uk (K.J.K.); michael.merchant@manchester.ac.uk (M.J.M.); 3The Christie NHS Foundation Trust, Manchester Academic Health Science Centre, Manchester M13 9PL, UK; 4Christie Medical Physics and Engineering, The Christie NHS Foundation Trust, Manchester M13 9PL, UK

**Keywords:** mechanistic modelling, modelling DNA repair, resection-dependent repair, DSB repair, NHEJ, slow kinetics, artemis

## Abstract

**Simple Summary:**

When cells are irradiated, their DNA can become damaged, this causes different types of repair processes to try and fix the DNA breaks. One of the most lethal types of DNA damage is double-strand breaks (DSBs). This work models the most used DSB repair process called Non-Homologous End Joining (NHEJ) and includes both its resection-independent and resection-dependent pathways. The models produced are benchmarked against experimental normal and deficient cell-types across a wide range of radiation qualities. We compare two approaches of modelling, the first is where the DSBs can repair in parallel and the second is where the DSBs repair is entwined. We find that it is necessary to consider both the resection-independent and resection-dependent pathways as entwined to produce a model which robustly matches experimental work. Through better modelling of NHEJ repair, it can improve our understanding of radiation response which has potential in biological optimisation for radiotherapy.

**Abstract:**

Mechanistic in silico models can provide insight into biological mechanisms and highlight uncertainties for experimental investigation. Radiation-induced double-strand breaks (DSBs) are known to be toxic lesions if not repaired correctly. Non-homologous end joining (NHEJ) is the major DSB-repair pathway available throughout the cell cycle and, recently, has been hypothesised to consist of a fast and slow component in G0/G1. The slow component has been shown to be resection-dependent, requiring the nuclease Artemis to function. However, the pathway is not yet fully understood. This study compares two hypothesised models, simulating the action of individual repair proteins on DSB ends in a step-by-step manner, enabling the modelling of both wild-type and protein-deficient cell systems. Performance is benchmarked against experimental data from 21 cell lines and 18 radiation qualities. A model where resection-dependent and independent pathways are entirely separated can only reproduce experimental repair kinetics with additional restraints on end motion and protein recruitment. However, a model where the pathways are entwined was found to effectively fit without needing additional mechanisms. It has been shown that DaMaRiS is a useful tool when analysing the connections between resection-dependent and independent NHEJ repair pathways and robustly matches with experimental results from several sources.

## 1. Introduction

Radiotherapy is the standard of care for many cancer treatment sites. The goal of radiotherapy treatment planning is to maximise the absorbed dose—a surrogate for biological effects—in the tumour volume, whilst sparing nearby healthy tissues. The fate of irradiated cells mainly depends on the generation of double-strand breaks (DSBs) and the accompanying DNA repair activation [1]. There are several repair pathways available for DSBs, the most dominant being Non-Homologous End Joining (NHEJ) which is a fast and efficient pathway for DSB repair. This process requires no homology between breaks nor the presence of a template from which to restore a break, and is available in all phases of the cell cycle [2]. The core components of NHEJ—the Ku70/80 heterodimer, DNA-Protein Kinase catalytic subunit (DNA-PKcs), XRCC4, and DNA ligase IV have been described in the literature [3]. Cells deficient in any of these components show increased radiosensitivity [4]. Whilst the core components of NHEJ have been well characterised, recent work has revealed the involvement of additional proteins and sub-pathways [5,6].

For over a century, mathematical models of experimental or clinical observations have been used to investigate potential mechanisms of biological responses following irradiation [7]. The linear-quadratic (LQ) model is the most widely used mathematical model for cell survival but has become empirical in nature, due to an oversimplification of the mechanistic processes described [8,9]. Although these earlier models were based largely on the experience of low Linear Energy Transfer (LET) irradiation (e.g., X-rays), models for proton and heavy ion therapy were also developed to meet the needs for emerging radiotherapy techniques. Many empirical models which apply simple modifications to LQ parameters have been developed to predict the relationship between Relative Biological Effectiveness (RBE) and LET for protons and carbon ions [10,11]. Such models have been successfully incorporated into clinical practice for carbon therapy to improve patient outcomes [10,12]. However, a major challenge in this field is the definition and quantification of the links between physics and biology. In an attempt to develop a mechanistic understanding of the underlying nature of radiation response, modelling DNA damage and repair at the cellular level has been a topic of great interest [13]. Some mathematical models have constructed DNA repair pathways by applying reaction kinetic equations to predict biological endpoints, although this approach tends to neglect the differences in initial DSBs from different irradiation qualities and does not adequately describe the variance in the repair functions between cell lines. More sophisticated models of radiation response are capable of predicting DNA damage complexity and repair fidelity through use of a track structure approach and mechanistic modelling of NHEJ and Homologous Recombination (HR) repair pathways [14,15,16,17,18,19]. Currently, the underlying nature of the slower component in the NHEJ repair process for G0/G1 phase cells is not explored using a mechanistic approach [6,20]. 

DSBs induced by irradiation have been observed to repair with biphasic kinetics in all phases of the cell cycle [21]. In late S and G2 phase cells, NHEJ attempts to repair most DSBs initially with processing by HR increasing at later time points [22], at least partially explaining the biphasic kinetics. However, HR is not available during G0/G1. There are other non-specific pathways such as microhomology-mediated end joining (MMEJ) and alternative end joining (alt-NHEJ), that involve microhomology (MH) usage. MMEJ usually occurs by Artemis-dependent NHEJ in G0/G1 phase human cells whilst alt-NHEJ is considered not to contribute significantly to DSB rejoining in G0/G1 cell phase [23]. However, analysis of DSB repair using pulse-field gel electrophoresis (PFGE) and γ-H2AX foci has revealed that in G0/G1 phase, around 85% of IR-induced DSB rejoining happens through a fast component (within 4 h) while a slow component accounts for the remaining 15% of DSBs [21]. The fast process in NHEJ has no requirement for resection-relevant factors whilst the slow process depends on DSB end resection [21,24]. Although proteins promoting resection in G0/G1 and S/G2 cell phases are similar, there are still important distinctions. In S/G2, RPA, Rad51, CtIP, BRCA1, and Rad52 proteins are key factors for HR [25]. The slow NHEJ process additionally requires the nuclease Artemis alongside other major NHEJ proteins (Ku70/80, DNA-PKcs, and ligation complex) [21,24]. This slow resection-dependent NHEJ initiates using CtBP-interacting protein (CtIP) and EXO1 proteins [24,26]. Furthermore, Ataxia telangiectasia mutated (ATM) appears to have a vital impact on the slow repair process by loosening the highly compacted chromatin [27] allowing subsequent repair proteins to be utilised.

Despite the current understanding of DNA DSB repair proteins involved in different repair pathways, the underlying temporal processing and combined action at DSB sites requires further probing. Mechanistic models for DNA repair have potential to facilitate the evaluation of biological uncertainty in radiotherapy treatment planning [7,28,29]. Recent work suggests there is a higher possibility of causing gene translocation formation, which can promote carcinogenesis, in the slow resection-dependent NHEJ process compared to the fast process. Biehs et al. showed that around half the number of translocations are generated in an Artemis-deficient cell line (CJ179) following 7 Gy X-ray radiation compared to wild-type 82-6 cells [24]. Furthermore, the increase in chromosome aberrations for G0/G1 cells via the NHEJ process has not been fully explained, although early studies have revealed a quadratic relation to dose [30].

This work aims to address the current gap in modelling the slow component of NHEJ DSB repair in the G0/G1 cell cycle phase using a Monte Carlo based method. Utilising the DNA Mechanistic Repair Simulator (DaMaRiS), within which the mechanisms arrived at by analysing in vivo/vitro experiments can be arranged and tested against known behaviours [18]. We investigate a slow, resection-dependent NHEJ repair process involving Artemis, CtIP, and EXO1 that interacts with the fast, canonical NHEJ repair process. We design and evaluate two structures of speculative repair pathways, A: “Parallel” repair pathway, and B: “Entwined” repair pathway. Validation of the models against experimental data is performed between wild-type and key protein deficient G0/G1 cell systems following radiation and the robustness of the model is tested. The current study provides a useful tool for exploring the overlaps and distinctions between the slow and fast processes. The models presented here are the first investigation of the slower NHEJ repair processes for normal tissue cells at G0/G1 phase.

## 2. Materials and Methods

The mechanistic modelling in this study uses DaMaRiS (DaMaRiS-v2020.06.25, The University of Manchester, Manchester, UK), a framework that has been developed at the University of Manchester [17,18,19]. The DaMaRiS framework is based on the Geant4-DNA toolkit, which extends the Monte Carlo radiation transport simulation capabilities to the DNA level [31].

### 2.1. DNA Damage Model

The DNA damage model used in this work is a well-established model developed by our group at the University of Manchester [17,18,19,32]. A 10 μm diameter nucleus at G0/G1 phase with chromatin structure defined by biological data from Hi-C experiments, as well as a model of the 30 nm chromatin fibre with individual bases and backbones has been used. This Hi-C data is a newly developed methodology to infer spatial conformations of the genome directly from experimentally measured genome contacts in the genome organisation field [33]. Track structure simulations of radiation have been applied to these geometries to determine the location, complexity, and density of DSBs for a given dose of a given radiation quality. A linear relationship is assumed between energy deposition within DNA volumes and DNA strand break (SB) induction probability, increasing from 0.0 at 5.0 eV to 1.0 at 37.5 eV [15]. SBs on opposite strands and separated by 10 or fewer base pairs are clustered to form a DSB site, otherwise they are scored as isolated damage sites. For the photon case, it is assumed that DSB yield follows a Poisson distribution with an average of 25 DSBs/Gy/cell. The DSBs are then randomly distributed throughout the cell nucleus, accounting for a higher degree of dose homogeneity relative to the proton case. The information on initial DSBs following exposure was scored in a Standard DNA Damage data format (SDD) commonly used for describing DNA damage from computational simulations [17,34]. Use of the SDD additionally allows tracking of the local chromatin environment of a DSB site. In this work we explored two methods of determining if the DSBs are in heterochromatin or euchromatin. Genomic location of DSBs determined from the Hi-C model was combined with Chip-Seq data to randomly assign breaks as heterochromatic or euchromatic based on the methylation status of the local area [33]. This results in 48% of breaks being heterochromatic [35]. Alternatively, a value of 25% heterochromatic breaks was taken from Löbrich et al. and chromatin status randomly assigned to achieve this figure [23].

### 2.2. DNA Repair Model; Workings, Assumptions, and Limitations

The DaMaRiS framework is a user-defined DNA repair model that can simulate the motion and reaction of DSB objects and repair proteins [18,19]. DaMaRiS parses the SDD from the damage model and separates each DSB site into two separate DBS ends, cleaving at the most proximal pair of SBs. These two DSB ends are then placed as separate objects in the simulation. Each individual DSB end contains information relevant to itself, such as additional backbone or base lesions, and relevant to the DSB site overall, such as the local chromatin environment.

Whilst initially co-located in space, the two ends from a DSB can move independently from each other through a Continuous Time Random Walk (CTRW) model of sub-diffusive motion which is implemented by having a trapped state where diffusion is disabled, as there are not yet any bridging proteins present to physically link them. Two DSB ends in proximity and with the appropriate proteins attached can form a synaptic complex. Two DSB ends that have formed a synaptic complex are treated as the same object and will move together. Two DSB ends in a synaptic complex may dissociate, breaking the protein–protein interaction bridging the two strands of DNA, at which point they are again treated independently. In this work we have not yet implemented any large-scale chromatin rearrangements which could move both ends of a DSB together, regardless of their physical attachment to each other. This type of motion, however, has less influence on the probability of two DSB ends meeting than their local motion, and as such has less impact on repair kinetics.

In DaMaRiS, DSB objects progresses through the user defined pathways by changing state in a step-by-step manner. For example, a DSB end may transition into a DSB end with Ku70/80 attached, representing the loading of that specific protein. These state transitions are governed by user defined time constants that represent the average time for that transition to occur, with actual transition times being drawn randomly for each individual DSB object. When a DSB object has multiple possible paths for progression, a random transition time is drawn for each possible state change and the shortest time is selected. In this manner, the repair pathway used is dictated by which proteins are recruited to the DSB end. Influence on pathway choice can then be modelled by linking specific factors to critical time constants; shortening them to increase likelihood or lengthening them to reduce the frequency of that pathway.

The reaction of a repair protein with a DSB end is a chemical process and as such the state progression of one DSB end has no influence on another, physically separate, DSB end. This is incorporated into the model by treating DSB end progression through a defined pathway independently to its partner end. To influence pathway choice for both ends of a DSB other factors must be considered. Both the chemical state of the DSB ends and local protein concentrations could be factors that influence pathway choice; however, these have not yet been implemented in DaMaRiS, but are important avenues for future investigations. In this work we have considered the possibility for local chromatin environment to dictate or influence pathway choice [23,36,37,38]. The level of local chromatin compaction around a DSB has been implemented as described above for the DNA Damage Model. In DaMaRiS this has the effect of altering the time constants for loading of CtIP and DNA-PKcs depending on heterochromatic or euchromatic status.

No experimental data has been found to fit to on differing recruitment kinetics dependent on chromatin status. As such, these are treated as free parameters in order to investigate possible influence. It is important to note that due to the lack of data, although we have modelled this as the influence of chromatin compaction, the true mechanism could be one of those discussed previously. As an extreme example, it could be that chromatin compaction has no effect on pathway choice and instead there is a proportion of complex breaks which dictate pathway choice that happens to be the same as the proportion of hetero-/euchromatin we have modelled. Models where chromatin compaction have not been included can be thought of as reporting on the average behaviour of protein recruitment in both heterochromatin and euchromatin.

### 2.3. Potential Models in G0/G1 Cells for Combined Slow and Fast Processes

The NHEJ repair model in this work consists of both the fast, resection-independent, and slow, resection-dependent, processes in G0/G1 normal tissue cells. We investigate the relative feasibility of two potential models for this pathway, a “Parallel” model (Figure 1A) where the two sub processes are kept separate, and an “entwined” model (Figure 1B).

One of the distinct features of the “*Parallel*” *repair pathway* (Model A) is that DSBs ends processing through the slow and fast repair process are treated separately in the whole process. That is, a DSB end handled by resection-dependent process cannot enter synapsis with another end handled by the fast process. However, in “*Entwined*” *repair pathway* (Model B), DSB ends resected by CtIP and EXO1 are capable of forming synaptic complexes with DSB ends handled by the fast process. Table 1 summarises the difference of key model parameters for the two scenarios.

**Table 1 cancers-13-02202-t001:** A summary of the model parameters and distinct features between two models. (Unit: seconds).

Process	Model A	Model B	Fitted
Ku70/80 Inhibition	0.85	0.85	Ref. [18]
Release from Ku70/70 Inhibition	3.8	3.8	Ref. [18]
Ku70/80 Recruitment	1.1	1.1	Figure 2a
DNA-PKcs Recruitment in fast process	1.2	1.2	Figure 2b
Artemis:DNA-PKcs Recruitment resection process	500.0	500.0	Figure 2b
CtIP Recruitment	7.0	7.0	Figure 2c
EXO1 Recruitment	1.2	1.2	Figure 2d
Become blunts	60.0	400.0	Figure 2e
Dissociation of synapsis	400.0	400.0	Figure 3 and Figure 4
Remove Base lesion	300.0	300.0	Ref. [18]
Remove SSBs	900.0	900.0	Ref. [18]
Stabilisation of synapsis	250.0	250.0	Ref. [18]
Ligation of two ends	1200.0/8000.0	3000.0	Ref. [18] and Figure 3 and Figure 4
Form synapsis (nm)	25	25	Ref. [18]
Jump diffusion coefficient (nm^2^·s^−1^)	6.0 × e^10^	6.0 × e^10^	Figure 5
Synapsis can form between DSB ends from different pathways	No	Yes	
Artemis & DNA-PKcs co-recruitment	Yes	Yes	
All DSB ends require Artemis	No	No	
Dissociation in slow process	No	Yes	
Final fixed ligation stage	Long (slow brach) and Fast (fast branch)	Intermediate	

**Figure 2 cancers-13-02202-f002:**
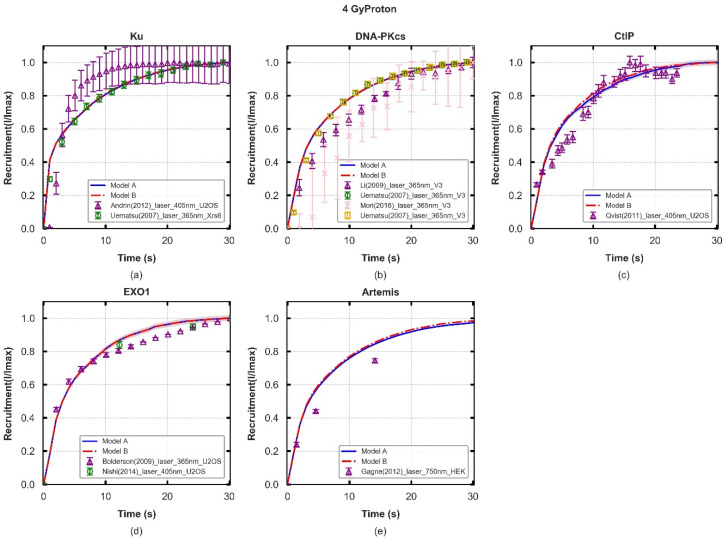
Protein fluorescence recruitment kinetics of two models following 4-Gy proton irradiation (lines) and experimental data (symbols), with experimental error bars showing the standard error of mean (SEM). The shaded areas are the SEM taken from 70 repeated simulations. Details of cell lines and exposure from the experiments are listed in Appendix A. Panel (**a**–**e**): values of Ku70/80, DNA-PKcs, CtIP, and EXO1 recruitment are normalised to the maximum during the 30 s and Artemis are normalised to the maximum value in 60 s. All values, for each protein, are normalised to the initial number DSBs for each repeat simulation and the averaged values are compared with experiments. The results for other irradiation types (0.5–4 Gy photon/proton) are collected in Appendix A.

**Figure 3 cancers-13-02202-f003:**
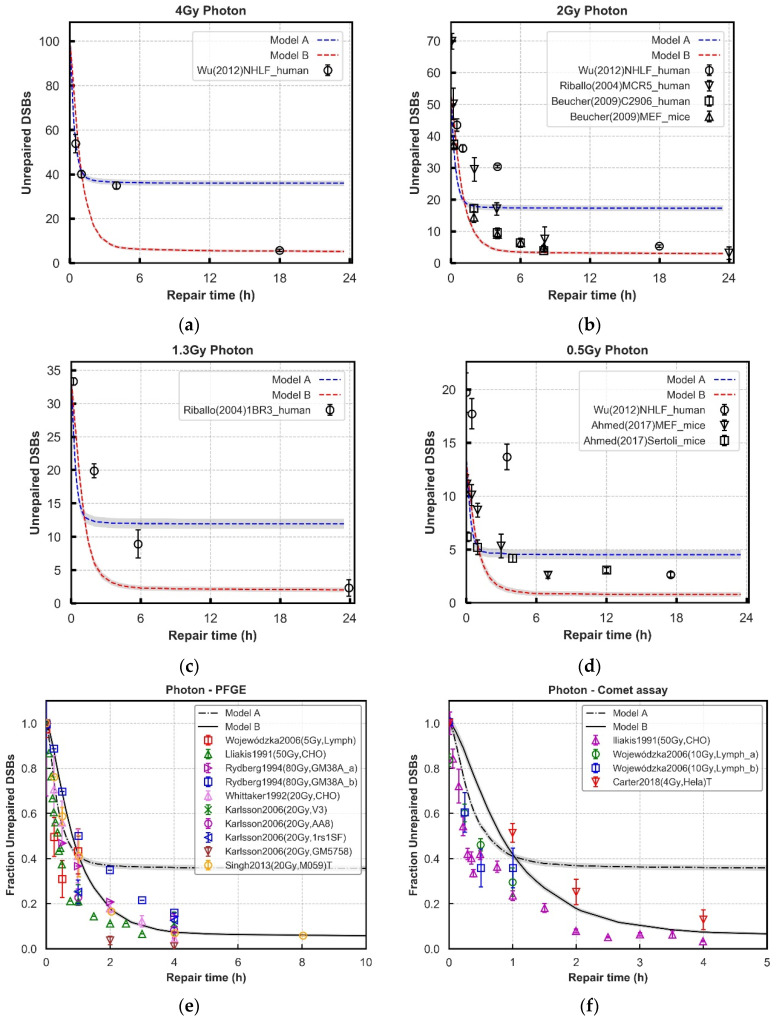
Comparison of repair kinetics between models and experimental observations in wild-type cell lines after 4, 2, 1.3, and 0.5 Gy photon (Panel (**a**–**d**)) exposure. The number of γ-H2AX or 53BP1 foci (black symbols) at various time points in NHLF, MCR-5, C2906, and 1BR3, MEFs and Sertoli cell are from experiments. Panel (**e**,**f**): Results from 4 Gy photon simulations are normalised for comparison against PFGE and comet assay literature data with a range of doses. The majority of cell lines are fibroblasts and lymphocytes. We use a label ‘T’ to specifically denote where the data is from tumor cell lines. All of the literature datasets are assumed or reported to be asynchronous, except Lliakis et al., which is synchronised in G1 cell cycle phase [51,52,53,54,55,56,57]. In all panels, lines with error bars representing SEM are simulated results from 70 repeats. Grey Shades represent error bars.

**Figure 4 cancers-13-02202-f004:**
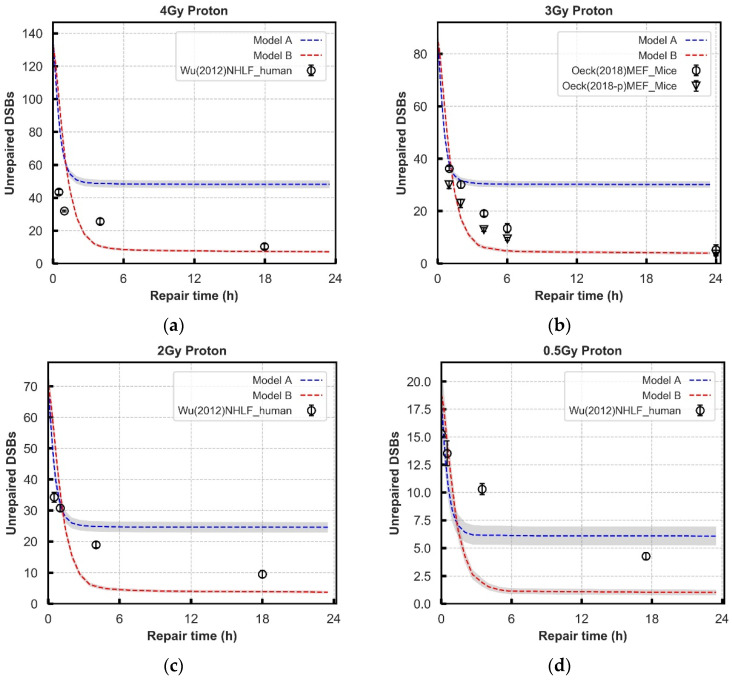
Comparison of repair kinetics between models and experimental observations in normal cell lines after 4, 3, 2, and 0.5 Gy proton (Panel: (**a**–**d**)) exposure. The average of 70 repeat simulations are shown by lines, with error bars representing SEM. The error bars are too small to be seen on this scale. The number of γ-H2AX foci (black symbols) at various time points are collected from experiments.

**Figure 5 cancers-13-02202-f005:**
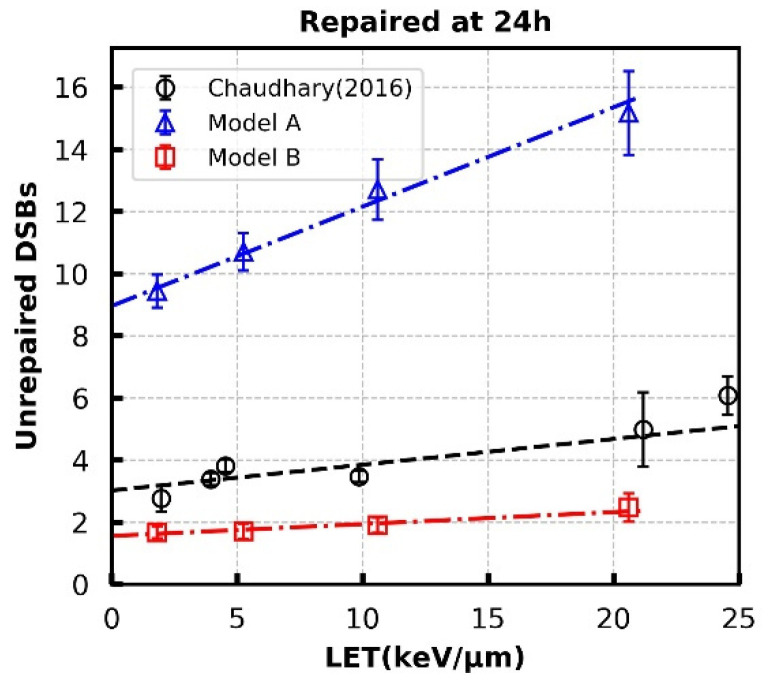
Unrepaired DSBs at 24 h for 1 Gy proton radiation at different LETs. Empty circles are results for normal human skin fibroblast (AG01522b) taken from Chaudhary et al. Error bars in simulated results (Model A and B) are SEM of 120 repeats for each data point. The dashed lines are linear-fits with the fitting parameters listed in Appendix A.

### 2.4. Model Evaluation; Fitting Data and Limitations

The experimental data selected to fit model parameters, or to assess model performance, has been limited to G0/G1 cell cycle phase. In this way we avoid confounding factors due to other cell cycle effects (such as Homologous Recombination). Time constants in the models, which govern the recruitment of repair proteins to DSB ends, are fitted to reproduce in vitro experiments from literature with cells irradiated by lasers [40,58,59,60,61,62,63,64]. We have discussed the suitability of applying laser induced damage repair kinetics to ionising radiation in previous work [18]. Additionally, recruitment data can only be used to infer loading kinetics and not protein action. Proteins may be recruited to sites through protein–protein reactions where their activation is superfluous to the processing of the damages or be recruited early for later use. However, logically, if a protein is not present it cannot act and as such this data is still useful for investigating the temporal action of repair factors in this manner.

The repair kinetics of DSBs for all proposed models are compared to γ-H2AX foci or 53BP1 foci experiments in literature with seven different G0/G1 phase fibroblast wild-type cells (NHLF, MCR5, HSF2, C2906, MEF, 1BR3, and Sertoli) and lymphocytes derived from healthy human volunteers exposed to doses of photons and protons between 0.5 and 4 Gy [21,65,66,67,68]. As these assays are indirect measures of DNA damage there are additional considerations. Firstly, their kinetics, although closely matched, are not directly that of the induction and repair of DNA strand damage. Secondly, due to geometry and the physics of light being processed through a microscope there is potential for both systemic under- and overcounting of foci which would alter the perceived repair kinetics, particularly at early time points [69,70]. Additionally, we also compare against PFGE and comet assay results from literature, which are a more direct measurement of DNA damage. We have normalised the all datasets, allowing for comparison against a range of radiation doses. This comparison to literature data includes a range of cell types, the majority of which are asynchronous.

Given the dependence on Artemis, the proposed models were evaluated against Artemis inhibited repair data. In DaMaRiS this can be simply achieved by removing time constants governing the loading of Artemis and leaving all other parameters the same. The Riballo data set is used to evaluate the agreement between DaMaRiS and experiment where two Artemis-deficient fibroblast cell lines (CJ179 and MEF) were exposed to 1.3 and 2 Gy photons [21]. The robustness of the models were additionally checked by constructing XLF-deficient pathways and comparing to literature repair data. XLF is not yet explicitly modelled in these pathways; however, it is commonly thought to contribute to the stability of synaptic complexes. To model this the dissociation time constant in both models was set as 11.0 s. Experimental data of XLF-deficient cells (2BN, HF) exposed to 2 Gy of photon radiation was extracted from Beucher et al. [67].

In these models, the quantity of unrepaired DSBs simulated at 24 h has a strong dependence on end mobility [18]. Therefore, the same parameter controlling motion is adopted to fit against the linear trend with LET in experimental data extracted from Chaudhary et al. [71]. This modelling work identifies the importance and scientific need for more data in this area, and we hope this may provide support for experimentalists in this field.

It should be noted that all experimental data used in this work are reproduced from published literature based on various cell lines. As such the parameters of the NHEJ pathway in this work is a general mechanistic description, and not an attempt to recreate the specific response of any one cell line exactly. If the hypothesised models can fit all data adequately, then this gives confidence in the general arrangement of mechanisms, which in reality will be perturbed by cell line specific factors.

The model’s ability to reproduce experimental results was assessed through reduced Chi-Square (χ^2^/DF). Model results are compared to each experimental data set separately, giving a number of Chi-Square statistics per protein. The final statistic is then quoted as the unweighted average of each Chi-Square, giving an overall impression of goodness-of-fit to the entire experimental data. Alongside this average, we quote the “best” and “worse” reduced Chi-Square, giving an impression of the model’s ability to fit an individual dataset. Given the previously mentioned limitations, the fit to literature data cannot conclusively determine which of the models tested represents reality, but an indication of likelihood can be given. Additionally, the models can be used to determine acceptable ranges for additional parameters that would increase or decrease confidence in a particular model, thus highlighting valuable future experimental tests.

## 3. Results

### 3.1. Recruitment Kinetics of Repair Proteins

To evaluate the potential repair pathways in G0/G1 phase cells, the protein recruitment kinetics of the models and experiments are compared and shown in Figure 2. The goodness of fit (χ^2^/DF) is summarised in Appendix A as well as the overall statistics of the “best” and “worse” fits (the smallest and biggest statistics respectively among all reproduced experimental data sets for each repair protein) to guide for further assessment for models’ behaviours.

There is a slight model difference between the recruitment kinetics for the responses of relatively high LET ionizing radiation and laser induced damage as shown in Appendix A, which is also reported previously by Warmhenhoven et al. [18]. For simulations performed in this work, no significant effect on initial protein recruitment kinetics from radiation modality or dose was found. The results for 4 Gy proton irradiation are represented in Figure 2, and the models have similar behaviours among the five repair proteins. For Ku recruitment, the model is compared to experimental data from Uematsu et al. [40], where Ku70/80 complimented Xrs6 cell lines from Chinese Hamster Ovary (CHO) were irradiated by a 365 nm pulsed nitrogen laser (Figure 2a). Here, we see good agreement in 30.0 s and around 80% recruitment is reached at 10.0 s for both models. None of the models fit well with Andrin’s experiments [61], which use UV laser micro-irradiated human osteosarcoma (U2OS). The experiments show faster recruitment of Ku than simulations and reach 80% of the maximum at 5.0 s. The simulated recruitment of DNA-PKcs considers all DNA-PKcs-related processes, both for fast and slow kinetics. Here, the simulated models fit better with 365 nm laser irradiated, DNA-PKcs complimented, V3 cell lines (belonging to CHO) [40] and reach 80% recruitment at 11.0 s. Ku and DNA-PKcs play a key role in both fast and slow NHEJ processes and the time constants in the faster phase are the same as those reported previously by Warmenhoven et al. [18].

Expanding on our previous work modelling the fast NHEJ repair pathway, we assume that CtIP is recruited after loading Ku (step B to C in Figure 1) and then EXO1 (step C to D) in the slow pathway [47]. The time constants are determined as 7.0 s and 1.2 s for CtIP and EXO1, respectively, to correctly fit the experimental recruitment behaviour. Therefore, the models have no difference within 30.0 s but have faster recruitment of CtIP in 10.0 s after exposure compared to the work of Qvist et al. [59] and slightly slower recruitment kinetics for EXO1 after 10.0 s when compared to Bolderson et al. using U2OS cell lines [58,64]. A relatively long (500.0 s) delay is introduced after EXO1 recruitment to account for the resection process before recruitment of DNA-PKcs:Artemis. When further investigating the recruitment of Artemis, good agreement is still achieved with models paired with experimental data points generated with 750 nm laser-irradiated Human Embryonic Kidney 293 cells (HEK 293) but with faster recruitment kinetics within 20.0 s. Around 80% recruitment of Artemis is completed at ~16.0 s. The modelled time constant for Artemis complex fulfilling the resection process, which produces blunt ends (step F to G in Figure 1) is set as 60.0 s (Model A) and 400.0 s (Model B) after multiple time constants for this process have been trialed and compared to experimental recruitment data. Final time constants for the dissociation stage are 400.0 s.

Taken together, this data shows that both hypothesised models can fit these recruitment data taken from the literature. To differentiate between the two models, additional endpoints must be considered, and therefore we investigate fits to repair kinetics.

### 3.2. Repair Kinetics for Wild-Type Cells

Further evaluation of the models is performed using experimental data of DNA repair kinetics in wild-type cells. To simultaneously fit this data and the recruitment data, the previously adjusted parameters are treated as fixed, substantially reducing the number of free parameters. The damage input used in the models is generated from the model published by Henthorn et al., simulating the same exposures reported in the experimental data [72]. In Figure 3 and Figure 4, the yields of unrepaired DSBs generated in the models are compared with the yields of γ-H2AX and 53BP1 foci in various cell lines from publications following different doses of photon and proton irradiation respectively at different time points, up to 24 h [21,66,67,68,73].

The “Parallel” process (Model A) has a fast repair rate in the first 1 to 2 h which fits well with most literature data, and then a plateau beyond 2 h where relatively more DSBs exist. The “Entwined” process (Model B) always produces less unrepaired DSBs than the “Parallel” process at longer repair time, which fits better with experimental data. However, with low-dose cases (Figure 3d and Figure 4c,d), it predicts a bit of higher (around 3~4) unrepaired DSBs with compared with experimental data sets. For the DSBs repaired in the first 2 h, both models have the best fit across all the tested data sets both for photon and proton irradiations. The detailed statistics in Figure 3 and Figure 4 for each exposure are shown in Appendix A. The final ligation time constants are 1200.0 s in the fast phase adopted from Warmenhoven et al. [18], and 3000.0 s for Model B in the slower process is a comprehensive decision of eight irradiations after fitting to experimental data. Appendix A shows how ligation time will change the repair results.

Whilst Model B tends to overestimate the repair rate for foci based assays, when considering the more direct measurements of damage (PFGE assay, and comet assay) the model tends to predict a slower repair rate compared to the experimental data (Figure 3e–f).

Furthermore, the relationship between the yield of unrepaired DSBs at 24 h following 1 Gy proton exposure across a range of LET for the models was investigated. As in Warmenhoven et al. [18] the main parameter affecting this end point is the scale of motion of individual DSB ends. Figure 5 shows the fit to experimental data with DSB end diffusion coefficients of 6.0 × 10^10^ nm^2^·s^−1^. Detailed linear fit parameters are given in Appendix A. The “Entwined” process (Model B) slightly under-predicts DSBs at 24 h but follows the same trend as experimental data. It should be emphasised that a large over-estimation of unrepaired DSBs appears in the “Parallel” process (Model A). Further investigation showed that the increase in unrepaired DSBs is a result of DSB ends progressing down mismatched sub processes, which in the “Parallel” process (Model A) are not allowed to form synaptic complexes with each other.

To investigate this further we considered if the chromatin compaction in the local area of the break could influence both ends of the DSB to take the same path in the “Parallel” process (Model A). At the extreme, forcing DSB ends located in heterochromatin to recruit CtIP and abolishing its recruitment in euchromatin, causes the model to fit repair kinetics for both 25% heterochromatin (Figure 6) and 48% heterochromatin. Logically, this does however lead to either 25% or 48% of breaks going through the resection-dependent process in contrast to the 15% predicted by Riballo et al. [21,35]. For the case of 25% heterochromatin, a preliminary investigation showed that a reduction in proportion of resection-dependent repair could be achieved by introducing a roughly 40% chance of recruiting DNA-PKcs in heterochromatin, whist maintaining a complete block of CtIP recruitment in euchromatin. This does however re-introduce a portion of breaks that cannot repair due to two DSB ends progressing down different processes. This increase in residual DSBs at 24 h can be compensated for in DaMaRiS if there is no contribution to isolation of DSB ends by the motion of DSB ends. This is not achievable with CTRW sub-diffusive motion [74] whilst maintaining the observed range of motion of DSB ends relative to each other [75]. For the 48% heterochromatin case, no combination of parameters could be found in this preliminary investigation that would produce 15% resection-dependent repair whilst maintaining acceptable residual breaks at 24 h.

### 3.3. Repair Kinetics for Protein-Deficient Cells

The “Entwined” model was further tested for rationality and robustness by comparing the kinetics of DSB repair of Artemis-deficient and XLF-deficient cells to literature reported experimental data.

The Artemis-deficient cell system and wild-type cell system have a similar repair rate within around 1–2 h both for experimental and simulated results of 2 Gy Photon irradiations (Figure 7a). At lower doses, 1.3 Gy, the simulated results overpredict repair to a degree compared to experimental results (Figure 7b). At the later repair time (after ~6 h), the Artemis-deficient cell system has more unrepaired DSBs compared to wild-type cell systems. Overall, in the Artemis-deficient experiments (four cell lines following 2 and 1.3 Gy photon IR) there is a similar behaviour displayed between experiment and model, with the model showing more exaggerated behaviour. In the XLF-deficient system (fibroblast 2BN following 2 Gy photon IR), the model has good agreement with experimental data points from Beucher et al. [21]. The repair rates in “Entwined” process are similar to experimental data distribution but exhibit a lag at the early 4 h after exposure to 2 Gy photons. The recruitment kinetics of proteins in these protein-deficient cell systems are displayed in Appendix A. The repair kinetics derivations are listed in Appendix A.

## 4. Discussions

In this work, two hypothesised NHEJ repair pathways, consisting of both resection-dependent and -independent processes, have been modelled. Model parameters were fitted to literature reported experimental data of protein recruitment, wild-type repair, and protein-deficient repair kinetics, as listed in Appendix A. It should be emphasised that the cell model constructed in this work is designed to represent generic normal-tissue cells, so the parameters in the repair model are not specific to any particular cell line. This work, therefore, should be viewed as investigating the general conserved mechanisms in NHEJ which could possibly be present across normal-tissue cells. Model parameters that produce good fits to experimental data could give some indications of the set of circumstances under which each model is more probable. These sets of parameters can then serve to focus further biological experiments, maximising their power to discriminate between the two models.

The initial investigation of repair protein recruitment kinetics (Figure 2), shows that both the “Parallel” and “Entwined” models are capable of reproducing literature reported experimental results. This data alone is therefore not enough to differentiate between the models. However, as the repair simulation is based on the recruitment of proteins, fixing their time constants here limits the number of free variables available to influence later repair. This is an important step in order to avoid over-fitting.

As discussed in methods, the recruitment data can only be used to identify the presence of repair proteins at a damage site, not their action. For example, no direct experimental data was found for completion of the resection process after EXO1 recruitment. As such, this was set at 500.0 s because a reduction in this time constant worsens the fit to CtIP and EXO1 recruitment (Appendix A). Included in this is the assumption that CtIP and EXO1 must dissociate from the DSB end in order for DNA-PKcs:Artemis to gain access and trim the single stranded overhang. These “action steps” in the model are therefore the parts we are least confident in, and where direct experimental data would be valuable. There is one limitation of utilising laser-induced DNA damage to validate the recruitment behaviour of the models. Laser induced DNA damage occurs by a not yet fully known mechanism as far as we are aware, although a possible mechanism could be thermal shock. We investigated if simulating 3000 breaks in a column of spot size of 1.7 μm^2^ produced different results of 7.5 MeV proton irradiation and found no difference (see Appendix A). Custom simulations designed to mimic laser irradiations were carried out and no significant difference was found in recruitment kinetics, although this drastically increased computational time.

After establishing the recruitment kinetics of all proteins in the models, the unrepaired DSBs are evaluated over 24 h following irradiation for a range of radiation qualities (Figure 3 and Figure 4). The experimental behaviour itself follows an overall trend, however, the combined literature data is noisy, which cannot be accounted for simply by cell type. For example, in Figure 3b there is more agreement between experimental data points from Beucher et al. for human and rodent fibroblast cell lines [67] than there is between the data from human fibroblast cell lines from Beucher et al. and Riballo et al. [21,67]. Models are adjusted to give a set of parameters that generally fit all the data, not to reproduce perfectly each individual experiment.

Neither model fits the experimental data perfectly, with the “Entwined” model B tending to overpredict repair and the “Parallel” model A underpredicting repair. The “Entwined” model B agrees reasonably well with experimental data at early time points. At intermediate time points the model tends to overpredict repair before levelling-off and agreeing well with experimental data at late time points. This may indicate missing mechanisms acting in the intermediate timeframe that we have not fully captured. At low doses of proton or photon irradiation this behaviour alters slightly in that the model overpredicts repair at late time points as well. This could be due to low-dose hypersensitivity which is not considered by our models [76]. When compared 1 Gy proton irradiations of various LET, model B only slightly underpredicts (2–3 breaks) the unrepaired breaks at 24 h, whilst following a very similar trend to the experimental data (Figure 5). Overall, the fit to data does not rule out that the general mechanism of NHEJ described in model B could be largely responsible for the observed experimental behaviour.

The “Parallel” model A predicts a higher number of unrepaired DSBs in wild-type cell system across later time points (Figure 3 and Figure 4). This is especially evident in Figure 5 where adjustment of the anomalous diffusion coefficient, generally the dominating factor in residual break yield when modelling CTRW sub-diffusive motion [74], could not allow the model to match experimental data from Chaudhary et al. [71]. This is due to the handling of individual DSBs by model A. For a DSB to be fixed in model A, both ends involved in forming a synaptic complex must have proceeded through the same pathway. As discussed in methods, DaMaRiS handles DSB ends independently whilst they are physically separate from each other. This results in each end randomly going through either the resection-dependent or -independent processes, resulting in a high likelihood of mismatched ends that cannot form synaptic complexes with each other. The fit to data of model A suggests that, in this state, the model is less likely to be a good explanation for observed experimental behaviour.

To correct for the mismatch probability in model A, an additional mechanism had to be included in the form of the heterochromatic or euchromatic state of the DSB. A good fit to the data was achievable when we considered the nucleus to be comprised of 25% heterochromatin with a 40% chance of resection-dependent repair in heterochromatin only. A plausible mechanism for this may be that the resection-independent process could be heavily favoured in the presence of an abundance of DNA-PKcs. If the more densely packed heterochromatin acted as a barrier to DNA-PKcs infiltration, it would give CtIP more of a chance to be recruited to DSB ends in these environments only.

Whilst this arrangement does result in the target 15% resection-dependent repair it still leaves a portion of DSB ends in a mismatched state. The extent of this mismatch in the simulation roughly matches the experimentally observed residual DSBs. This then requires that the motion of the DSB ends make no contribution to the residual DSBs. In DaMaRiS, CTRW sub-diffusion is implemented as bursts of movement, or “jumps”, separated by random lengths of time when motion is halted. One of the behaviours that result from this is that objects are less likely to revisit locations. If the magnitude of the “jumps” are sufficiently reduced then, even though the DSB ends would not likely revisit the same exact location, they may still remain local enough to each other to form a synaptic complex. However, this reduced movement contradicts some experimental evidence showing movement of DSB ends relative to each other in the order of hundreds of nanometres [41,75]. One solution to this could be an alternate form of sub-diffusive motion as suggested by Lucas et al. [77], implemented as a visco-elastic boundary. This bounded motion would allow for initial separation of the DSB ends as observed by Soutoglou et al. [75], whilst ensuring that the ends will encounter each other at longer time points, thus not contributing to residual breaks.

Therefore, for the “Parallel” model A to be as likely as the “Entwined” model B, both the additional mechanism of chromatin compaction influence of protein recruitment, and an alternate model for DSB end sub-diffusive motion would be required. Both of these elements have support in the literature, but we are unaware of definitive proof; serving to highlight these points as areas where experimental investigation would have an important contribution to further understanding the NHEJ pathway.

Alternatively, a certain chemical structure of DSB end could exist that encourages the recruitment of CtIP over DNA-PKcs. If this occurred with a 15% probability, then it could be the underlying factor in resection-dependent repair, as expected in some publications [23]. However, it must be considered that the attack on opposite strands of DNA are often from independent processes in a radiation track [78]. If this behaviour is driven by, for example, a specific 5′ structure, this would again lead to the mis-match problem. Speculatively, the behaviour could instead be driven by a specific combination of 5′ and 3′ structures on the same strand that could both result from a specific chemical attack.

Further investigation of the robustness of the “Entwined” model B was carried out through simulation of literature reported experimental data with key proteins knocked out or inhibited. As DaMaRiS is comprised of discrete steps linked together, deficiencies are easy to model by altering specific links in the model whilst leaving all other parameters unchanged. In the model, Artemis is included in both the resection-dependent and -independent processes due to the assumption that it is recruited with DNA-PKcs. However, when progressing through the resection-independent pathway, only the presence of DNA-PKcs is required. Artemis knockout therefore has no effect on this pathway in the model. However, the activation of Artemis could still be required if there is a large single-strand DNA overhang needing to be cut [23,42], a mechanism which is currently missing from model B. In the resection dependent process Artemis has been experimentally identified as a key component [79] and its loss results in an increase of unrepaired DSBs [23]. In the model this happens as all DSBs that recruit CtIP will end up blocked from progressing to blunt DSB ends due to the lack of Artemis. The model overpredicts the impact of Artemis knock-out, perhaps indicating the presence of a corrective mechanism in the nucleus that rescues a portion of the breaks stalled in resection dependent repair. Both experimentally [24] and in simulations of model B it has been shown that this repair defect can be rescued by further inhibition of CtIP (Appendix A).

Further evaluation against the XLF-deficient cell lines was performed. XLF protein availability will affect the final steps of NHEJ repair possibly by providing additional stability to Lig4 and XRCC4 [5], but its function is underexplored. Therefore, to model XLF deficiency in model B, progression to stable synapsis was weakened by decreasing the dissociation time in models. This increases the likelihood that synaptic complexes will dissociate into naked DSB ends. As no direct experimental data could be found, the time constant was varied to find the best fit to repair kinetics. With a time constant for dissociation of 11 s, model B can reproduce the repair trend compared with fibroblast cells (2BN) [67] but with a slower repair rate at the earlier times.

We compared the performance of model B to other in silico models of DNA repair [14,20,80], and show good agreement with these models, with the caveat that the modelled repair pathways are different and do not include modelling of resection-dependent and -independent NHEJ. Appendix A describes these differences.

The above results gives good confidence that the general mechanism described in the “Entwined” model B could be descriptive of most influential biological mechanisms at play, although not capturing all mechanisms that influence NHEJ repair. There are still some simplifications for resection-processing steps present in our model. For example, we neglect explicit description of the ATM protein, which activates several signalling pathways after DNA damage. These signalling factors have important effects on the slow repair process, such as association with the mediator proteins (e.g., Artemis). Additionally, ATM is suggested to have a role in loosening highly-compacted chromatin during repair [23], which may imply greater mobility for DSB end transportation and higher yields of mis-repair.

## 5. Conclusions

DNA repair has a close relationship with cell survival and NHEJ is the main repair pathway, so understanding the temporal functionality of key NHEJ proteins is crucial. Why certain breaks undergo the resection route, when no HR is available, is still an open question worthy of further investigation.

This study has investigated the NHEJ DSB repair pathway within a mechanistic model for the G0/G1 phase of the cell cycle following irradiation. The parameters used in the models are verified by experimental observations taken from the literature. The “Entwined” model B could match the experimental results from several sources for protein recruitment kinetics and overall repair kinetics, in both wild type and specific protein deficient cell lines. For the “Parallel” model A to fit experimental results additional mechanisms had to be considered. This gives less confidence in model A being representative of the true biological mechanisms but does highlight specific areas of investigation that could support it.

For this reason we favour the “Entwined” description of the pathway and propose that it is more likely that the resection-dependent NHEJ process could be considered a corrective mechanism for getting double strand breaks back into the NHEJ pathway, rather than as a separate pathway altogether.

## Figures and Tables

**Figure 1 cancers-13-02202-f001:**
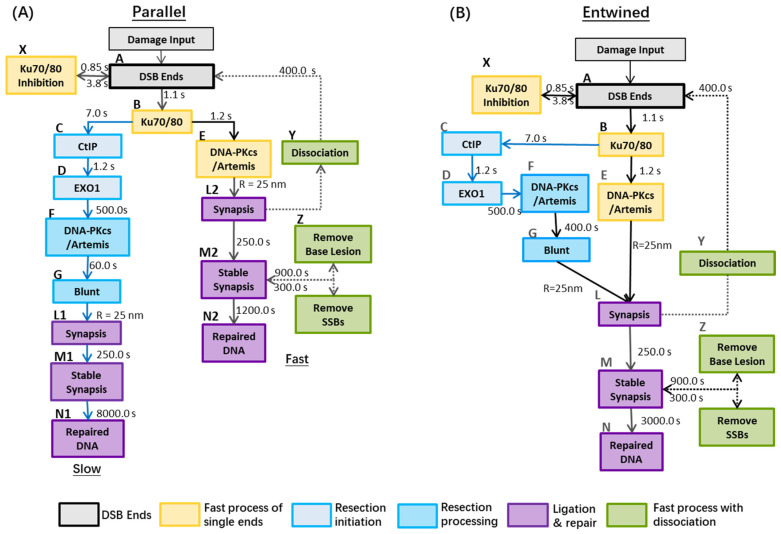
Repair models with slow and fast processes: Panel (**A**): (Parallel) There are no overlaps between the fast and slow repair kinetics after Ku recruited to DNA damage site. It is assumed, in the resection process, that there is no dissociation of synapsis to DBS ends and no clearance of base lesion and SSBs. Panel (**B**): (Entwined) DSBs from either the fast process or slow process (e.g., DSB ends handled by CtIP and EXO1) can form synaptic complexes with each other, then end-processing by Artemis and other proteins (e.g., DNA-PKcs:Artemis complex) can occur before final ligation. Reference: A to B (Lee et al. [39]); B to C (Barton et al. [26] & Löbrich et al. [23]); B to E (Uematsu et al. [40]); C to D (Biehs et al. [24]); E to F to G (Riballo et al., & Biehs et al., & Löbrich et al. [21,23,24]); L to N (Graham et al. [41]).The models are initialised by the damage input of DSBs, where each DSB is comprised of two ends. In both models, the binding of the Ku70/80 heterodimer to a DSB end occurs within seconds and its resultant complex serves as a platform for the subsequent loading of NHEJ proteins [39,42,43,44]. In resection-independent repair, DNA-PKcs is recruited to confer end protection forming the DNA-PK complex. DNA-PK autophosphorylation then facilities subsequent NHEJ factors such as the ligation proteins [45]. In resection-dependent repair, following the inward translocation of Ku [23], the resection–initialisation process takes place by MRE11 exonuclease (for 3″-5′ ends) or by EXO1 (for 5′-3′ ends) [23,46] and is also regulated by CtIP, which is suggested to promote the initiation of resection [24,26]. These processes are simplified in the model to recruit only EXO1 and CtIP, respectively [47]. Ku70/80 remains bound to the DSB end during this process [24,48] and helps bind DNA-PKcs to the DSB end post-resection, forming the DNA-PK complex [49]. DNA-PKcs facilitates recruitment of Artemis and stimulates its activity, which plays a key role in terminating the resection process [21,45,48,50]. In all cases, we assume that Artemis and DNA-PKcs are recruited together [48]. DNA-PKcs autophosphorylation gives Artemis access to DNA ends to cleave the single stranded DNA and produce a blunt-ended DSB [3,21]. These DSB ends are prepared to re-join with each other to produce a synaptic complex, facilitated by the DNA-PK complex. At the end of the model, stable synapsis is produced by the enzymatic activity of ligation proteins before final repair [41,42].

**Figure 6 cancers-13-02202-f006:**
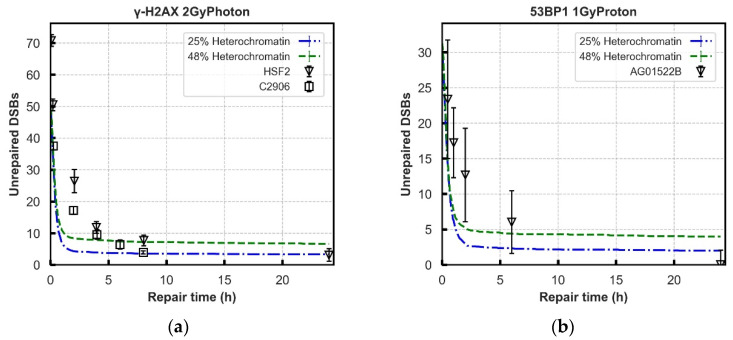
The repair kinetics of “Parallel” pathway (Model A) assigned with 25% and 48% heterochromatin (forced to recruit DNA-PKcs only, and euchromatin is forced to recruit CtIP) under: Panel (**a**): 2 Gy photon and Panel (**b**): 1 Gy 1.7 keV/um proton irradiation. The recruitment kinetics of proteins in these scenarios are displayed in Appendix A.

**Figure 7 cancers-13-02202-f007:**
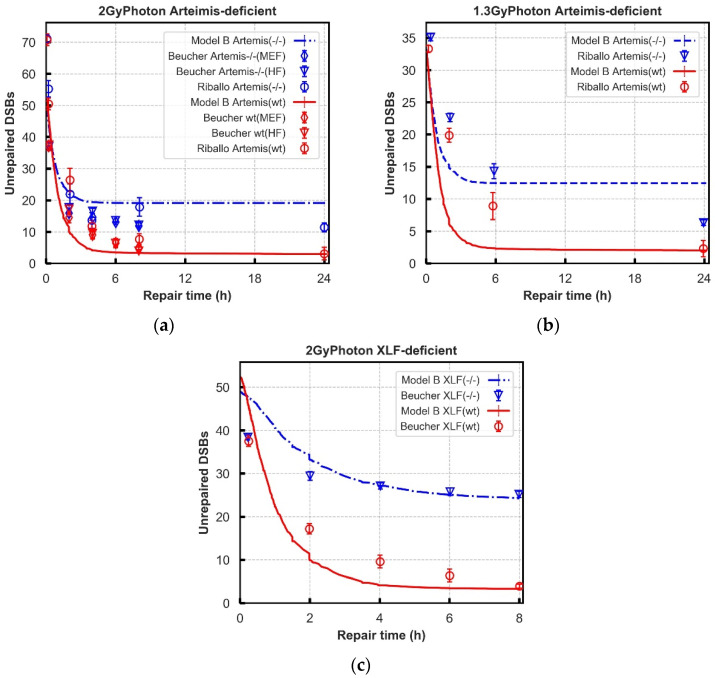
Comparison of unrepaired DSBs in Model B (lines) with the number of γ-H2AX foci obtained from experiments (symbols). (Panel: (**a**–**c**): Artemis-deficient and wild-type cell lines exposed to 2 Gy X-rays [45,67] and 1.3 Gy photons from ^137^Cs source [21]; Panel (**c**): XLF-deficient and wild-type cell line exposed by 2 Gy X-rays [67].

## Data Availability

The data presented in this study are available on request from the corresponding author.

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
