# Peer review of "Mechanistic Modelling of Slow and Fast NHEJ DNA Repair Pathways Following Radiation for G0/G1 Normal Tissue Cells"

_cancers, 2021, doi:10.3390/cancers13092202_

Round 1

Reviewer 1 Report

Authors should throughly revise the sentences in Page 2, Line 38-42, which have grammatical errors and typos.

Author Response

We apologise for the mistake. We have corrected this on Page 2 Line 38-42.

Reviewer 2 Report

The quality of the manuscript has been improved significantly after revision. I do not have any major concerns.

Author Response

We appreciate your comments.

Reviewer 3 Report

I appreciate the authors responses to my comments. However, and in my opinion, the manuscript would still be more suitable for a specialised journal rather than Cancers. Additionally, the major comments relating to lack of experimental data to fit the modelling data to (vitally important), cell type variability, and use of indirect markers of DNA double strand breaks are still weaknesses of the manuscript which have yet to be resolved.

Reviewer 4 Report

The authors have addressed all concerns.

Author Response

We appreciate your comments.

Round 2

Reviewer 3 Report

On reflection, and having read through the latest revisions made by the authors, I can agree to accept the article in its current form.

This manuscript is a resubmission of an earlier submission. The following is a list of the peer review reports and author responses from that submission.

Round 1

Reviewer 1 Report

In the current manuscript, authors have tested two in silico models for DNA double strand break repair in G0/G1 phase of the cell cycle using a previously developed tool DaMaRiS.  They have demonstrated that the model where slow and fast repair pathways are "entwined" fits better than the "parallel" model with the published reports of DSB recruitment kinetics of the DNA repair factors. As they authors have acknowledged these in silico models are overly simplistic and fails short to include a plethora of in vivo parameters, signaling pathways, and other repair factors. Although, the matching of the entwined model with the experimental data deserves merit, the current work is incremental.

In some figures authors have considered several data sets, while in others only one data set for matching. Authors should include at least two data sets in figure 3 (a,c), 4 (a, c,d). 

Minor comments:

1) Define DaMaRiS and NHEJ in introduction

2) Discuss about alternative end joining/ MMEJ in introduction

3) Define HiC

4) Page 3, line 36: What type of "radiation" ?; line43: What type of "ion"?

5) Page6, line 7: Both endonuclease and exonuclease activity of Mre11 is required to initiate resection

6) Define CTRW

7) Thoroughly check for typos.

Reviewer 2 Report

Previously, the authors constructed a computational framework DaMaRis to simulate the DNA repair process in silico. In this work, the authors extended their previous work to study the NHEJ repair mechanism by comparison of the simulated data from two plausible repair pathways and the experimental data. By examining data of protein recruitment kinetics, repair kinetics and unrepaired DSBs at 24 h, the authors found that the “Entwined” model B would match better with the experimental data. Although the “Parallel” model A could not be ruled out, it would require additional mechanisms such as chromatin states to be considered in the model. The authors concluded that model B would be the preferred model from the current experimental data.

The manuscript is within the scope of Cancers and would be interesting for both biologists studying DNA repair and computational scientists modeling biological systems. The manuscript seems mostly sound and is written in a clear language. The background information and references selection is largely adequate. However, I think the authors needs to check the manuscripts more carefully, as there seems to be a few inconsistency, and in a few places the results need more explanation.

I would like the authors to consider the following points.

  1. On page 11 line 19, the authors fit the experimental data with DSB end diffusion coefficient 6×1010 nm2/s, which equals to 6×10-8 m2/s. The authors need to check this number, as it is large than the diffusion coefficient of sodium ion in water (~1×10-9 m2/s). The diffusion coefficient for a protein should be ~1×10-11 m2/s or lower.
  2. In Table A3, the R2 for the experimental data did not look correct. Please double check calculation.
  3. In Table A4, it would be a good idea to mention whether the cells were synchronized in the experiments. In the models used in this manuscript, the HR pathway was not considered, which was a main limitation of the manuscript. If the cells were synchronized in the experiments, it was possible that the cells might in S/G2/M phase and were significantly different from the simulation condition.
  4. In the legend of Figure A1, the authors used “Figure S2”. It seems to be a mistake.
  5. In Figure 2 and A1, the experimental data look the same with different radiation doses. Is this the case or not?
  6. In the legend of Figure A2, it was referred to Figure 6. But from the text, it should be Figure 7.
  7. In the legend of Figure A4, it was referred to Figure 7. This figure was not mentioned in the text. It seems that it should be referred to Figure 6.
  8. Inside of Figure A6, the label mentioned “Model C”. Was this a typo?
  9. On page 11 line 10, the authors stated that the model B predicted higher unrepaired DSBs with low-dose cases. But the Figure 3 and 4 showed that model B had lower unrepaired DSBs. Please clarify it.

Reviewer 3 Report

This manuscript is centred on using mechanistic in silico models to determine events occurring during non-homologous end-joining repair of DNA double strand breaks in normal cells within resting phase (G0/G1) of the cell cycle. Using a Monte Carlo based model developed by the same Group, two models (parallel and entwined) are examined in terms of protein recruitment (of Ku, DNA-Pkcs, CtIP, EXO1 and Artemis) and double strand break repair kinetics (γH2AX foci), against reported experimental data using laser irradiation and following photon/proton irradiation, respectively. The major findings are that the models fit reasonably well with repair protein recruitment data, but have some deficiencies when examining repair kinetics through γH2AX foci as a surrogate marker of double strand breaks. The models also have some difficulties predicting unrepaired strand breaks as a function of proton LET (comparable against only a single study). Use of the entwined model additionally appeared to show discrepancies against experimental data in repair-deficient (Artemis and XLF) cells.

My major comment is that the manuscript is very mechanistic in nature and would be more suitable to a specialised DNA repair journal, rather than the readership of Cancers related to cancer and cancer biology. This is particularly pertinent given the data is centred on modelling DNA repair in normal cells. A few more specific comments are included below.

  • Recruitment of proteins is modelled against data acquired following laser-induced DNA damage. This is obviously not ideal given that laser damage will consists of DNA strand breaks with (mainly) or without modified ends, which is mentioned in brief in the Methods. More mention of the limitations of utilising this data should at least be built into the Discussion.
  • The two models obviously show some deficiencies in predicting DNA repair deficiencies through comparative analysis with γH2AX foci data. There is some correct discussion about this related to chromatin compaction, but this will also relate to the relative competency and efficiency for cellular repair (both elements which will differ from cell to cell) that the models don’t accommodate for.
  • A major issue, which is partially mentioned in the Methods section, is that γH2AX (and 53BP1 or RAD51) foci are indirect markers of DNA double strand breaks. Therefore, experimental data using this end-point do not reflect the presence of the DNA damage itself. Simulations should ideally be compared against direct DNA damage measurements, such as from comet or PFGE analysis.
  • Another limitation is the number of experiments which the modelling is compared against. In instances this is just a single dataset (such as 1 and 4 Gy photons in Figure 3, 0.5-4 Gy protons in Figure 4, plus LET-comparative data in Figure 5). It is therefore very difficult to draw accurate conclusions about model fitting from such data reported in single studies. A major suggestion here would be to generate additional experimental data, if the laboratory has capacity for measuring DNA double strand break levels in normal cells following photons and protons (or could be produced in collaboration).

Reviewer 4 Report

The manuscript by Qi et al. is extremely well written. The topic of the research presented is relevant to the field of radiation biology for improved understanding of the mechanism of DNA repair. 

Comments for minor revision:

1) In the Introduction, the authors state "...this approach tends to neglect the differences in initial DSBs from different irradiation qualities and do[es] not adequately describe the variance in the repair functions between cell line." (Line 27-28).

2) In the Methods, Table 1 lays out the times inputted into the two models required for protein activation and recruitment as well as for the enzymatic reaction. The major differences between Model A and B are the time for forming the blunt ended DNA on the DSB and the time for the ligation of the two ends. The authors need to spend some more time in the Methods and/or the Results to describe how these numbers were obtained for the calculations. Also, for rates that were actually measured, the authors can emphasize that these are actual data.

3) For the final ligation times inputted into the two models (lines 14-15, pg 11), it would be helpful if the authors could show the progress for the line fitting, using different numbers for the ligation times. This data could be placed in the Supplemental data, to show how the final numbers were achieved for each model.

3) The authors should spend some time in the Discussion to explain the limitations of their model, especially given that some of the rates selected for the model have never been measured in vivo.  Are there any similar models that have been previously proposed for DNA repair kinetics?